**Research**

# Epidemiological and clinical features of human metapneumovirus in hospitalised paediatric patients with acute respiratory illness: a cross-sectional study in Southern China, from 2013 to 2016

Ling Zhang,[1] Wenkuan Liu,[1] Donglan Liu,[1] Dehui Chen,[2] Weiping Tan,[3] Shuyan Qiu,[1] Duo Xu,[1] Xiao Li,[1] Tiantian Liu,[1] Rong Zhou[1]

LZ, WL and DL contributed equally.

For numbered affiliations see end of article.

**Correspondence to**
Dr Rong Zhou;
zhourong@gird.cn

## ABSTRACT

**Objectives** Human metapneumovirus (HMPV) is one of the most important respiratory viral pathogens affecting infants and children worldwide. Our study describes the epidemiological and clinical characteristics of HMPV present in patients hospitalised with acute respiratory illness (ARI) in Guangzhou, Southern China.

**Study design** A cross-sectional study.

**Setting** Two tertiary hospitals in Guangzhou.

**Participants and methods** Throat swabs were collected over a 3-year period from 5133 paediatric patients (≤14 years) hospitalised with ARI. Patients who are HMPV positive with clinical presentations (101/103) were recorded for further analysis.

**Results** Of the 5133 patients included in the study, 103 (2.0%) were positive for HMPV. HMPV was more prevalent in children ≤5 years (2.2%, 98/4399) compared with older children (>5–14 years) (0.7%, 5/734) (P=0.004). Two seasonal HMPV peaks were observed each year and mainly occurred in spring and early summer. Overall, 18.4% (19/103) of patients who are HMPV positive were codetected with other pathogens, most frequently respiratory syncytial virus (36.8%, 7/19). Patients who are HMPV positive presented with a wide spectrum of clinical features, including cough (100.0%, 101/101), abnormal pulmonary breath sound (91.1%, 92/101), fever (88.1%, 89/101), expectoration (77.2%, 78/101), coryza (50.5%, 51/101) and wheezing (46.5%, 47/101). The main diagnosis of patients who are HMPV positive was bronchopneumonia (66.7%, 56/84). Fever (≥38°C) (91.6%, 76/83) was detected more often in patients with only HMPV detected than in patients with HMPV plus other pathogen(s) detected (72.2%, 13/18) (P=0.037), whereas diarrhoea was more common in patients with HMPV plus other pathogen(s) detected (22.2%, 4/18), compared with patients with HMPV only (3.6%, 3/83) (P=0.018).

**Conclusions** HMPV is an important respiratory pathogen in children with ARI in Guangzhou, particularly in children ≤5 years old. HMPV has a seasonal variation. Bronchopneumonia is a major diagnosis in patients who are HMPV positive.

## Strengths and limitations of this study

► Five thousand one hundred and thirty-three patients hospitalised with acute respiratory illness at two large municipal tertiary hospitals (hospital beds >1000) were enrolled on the study over 3 years.

► Patients aged from 1 day to 14 years old were tested for human metapneumovirus (HMPV) using TaqMan real-time PCR.

► Clinical characteristics of patients with HMPV positive were recorded.

► There was incomplete HMPV copathogen detection, because bacterial pathogens were not detected for patients who are HMPV positive.

► The clinical characteristics of patients who are HMPV negative and outcome data (discharge/death) for the patients who are HMPV positive were not collected for further analysis, limiting our understanding of this pathogen.

## INTRODUCTION

Human metapneumovirus (HMPV) is a non-segmented, negative-sense single-stranded RNA virus, which belongs to the Paramyxoviridae family.[1] HMPV was first discovered in 2001 in the Netherlands, after being isolated from a paediatric patient with acute respiratory illness (ARI).[2] Since then, HMPV has been associated with acute respiratory disease in individuals of all ages worldwide. Children, elderly and immunocompromised adults are most at risk of contracting the virus.[3–5] Children younger than 5 years of age seem to be particularly susceptible to HMPV. Previous studies,[6–12] which were mainly conducted in developed countries, have revealed that approximately 2.5%–11.3% of respiratory samples were

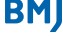

positive for HMPV in children ≤5 years, and 90% of individuals were seropositive for HMPV by 5 years of age.[13]

HMPV causes a variety of clinical symptoms ranging from a mild upper respiratory tract infection (URTI) to life-threatening lower respiratory tract infection (LRTI).[3 14–16] However, to date, there is no effective vaccine or specific medication either for the prevention or treatment of HMPV infection. Consequently, it is imperative to conduct more studies, especially in low/middle-income countries, to understand this pathogen in different areas and populations.

In this study, we investigated the epidemiological and clinical features of HMPV in paediatric patients, from July 2013 to June 2016. The findings of this study will help understand the distribution of HMPV in a subtropical region. Our results also provide a valuable insight into the clinical features of HMPV, which will improve early clinical diagnosis.

## METHODS
### Study design and respiratory sample collection
We conducted a cross-sectional study at two tertiary hospitals in Guangzhou, Southern China. Throat swab samples (n=5133), from paediatric patients (≤14 years) hospitalised with ARI, were collected at two hospitals between July 2013 and June 2016. ARI was defined as an illness that presented with at least two of the following clinical presentations: cough, nasal obstruction, coryza, sneeze and dyspnoea during the previous week. Patients, who were diagnosed with pneumonia by chest radiography during the previous week, were also included in the study, even if they did not show the clinical features described above. Some patients, who had been cured and discharged some time ago and readmitted because of new episodes of ARI, if met the recruitment criteria, were included in the study as new cases, otherwise excluded. Chest radiography was performed based on the clinical situation of the patients. The samples were collected according to established clinical protocols.[17] The samples were refrigerated at 2°C–8°C in viral transport medium, transported on ice to the State Key Laboratory of Respiratory Diseases and analysed immediately or stored at −80°C before analysis as previously described.[18]

The patients' clinical presentations or diagnoses were recorded from patients' medical records by attending physicians, using designed presentation cards and were categorised retrospectively into the following four groups: URTI, LRTI, systemic influenza-like symptoms and gastrointestinal illness. Patients with nasal obstruction, coryza, sneezing, coughing, expectoration or hoarseness were categorised as having URTI. Patients with bronchiolitis, pneumonia, bronchopneumonia, increased lung markings, dyspnoea or an abnormal pulmonary breath sound were categorised as having LRTI. Patients with a high fever (≥38°C), chills or debilitation were categorised as having systemic influenza-like symptoms. Patients with vomiting, poor appetite or diarrhoea were categorised

as having gastrointestinal illness. Some patients were assigned to several clinical presentation groups. Patients with incomplete clinical data were excluded from this analysis. Increased lung markings, bronchopneumonia, pneumonia and bronchiolitis were diagnosed by chest radiography. Abnormal pulmonary breath sounds included phlegmatic, wheezy, bubbling and moist rales. Other clinical symptoms were identified by a general medical examination and clinical descriptions, as previously reported.[19]

### Real-time PCR for HMPV detection
RNA was extracted from the throat swab samples with the QIAamp Viral RNA Mini Kit (Qiagen, Shanghai, China), according to the manufacturer's protocols. HMPV was identified by using TaqMan real-time PCR assays, as previously reported,[18] using kits from Guangzhou HuYanSuo Medical Technology according to the manufacturer's protocols. In brief, 50 µL RNA were extracted from a 200 µL sample, and real-time PCR was conducted using 25 µL reaction mix, containing Moloney murine leukemia virus reverse transcriptase, Taq polymerase and 5 µL extracted RNA. Cycling conditions included an initial reverse transcription at 55°C for 10 min, incubation at 94°C for 2 min, followed by 40 cycles of 94°C for 10 s and 55°C for 35 s (ABI-7500 real-time PCR instrument; Life Technologies, Singapore).

### Detection of common respiratory pathogens in patients who are HMPV positive
HMPV-positive samples were tested simultaneously using TaqMan real-time PCR assays for the following 17 respiratory pathogens: respiratory syncytial virus (RSV); parainfluenza virus types 1–4 (PIV1–4); influenza A and B viruses (InfA, InfB); adenovirus (ADV); enterovirus (EV); human coronaviruses (HCoV-229E, HCoV-OC43, HCoV-NL63 and HCoV-HKU1); human rhinovirus (HRV); human bocavirus (HBoV); *Mycoplasma pneumoniae* (MP); and *Chlamydophila pneumoniae* (CP). The testing procedure was conducted using kits from Guangzhou HuYanSuo Medical Technology as previously described.[18]

### Statistical analysis
All statistical analyses were performed with SPSS statistical software (V.19.0). To compare categorical data, $X^2$ and Fisher's exact tests were used, as appropriate. All tests were two-tailed, and $P < 0.05$ was considered statistically significant.

## RESULTS
### Detection of HMPV from patients with ARI
In total, 5133 paediatric patients, ranging from 1 day to 14 years old were enrolled in this study. The male to female ratio was 1.75 (3269:1864) and the median age was 33 months (IQR 9–48 months). Of the 5133 patients, 103 (2.0%) were positive for HMPV. The male to female ratio was 1.86:1 (67:36) in patients who are HMPV positive and

**Table 1** Distribution of copathogens in 19 patients who are HMPV positive

| Copathogens* | No of patients |
| --- | --- |
| RSV/HMPV | 4 |
| HCoV-OC43/HMPV | 3 |
| ADV/HMPV | 2 |
| HRV/HMPV | 2 |
| MP/HMPV | 1 |
| PIV2/HMPV | 1 |
| PIV3/HMPV | 1 |
| RSV/MP/HMPV | 1 |
| RSV/InfA/HMPV | 1 |
| RSV/HBoV/HMPV | 1 |
| MP/HBoV/HMPV | 1 |
| InfA/HBoV/HMPV | 1 |

*Patients who are HMPV positive were tested for 17 common respiratory pathogens.
Influenza B viruses, parainfluenza virus types 1 and 4, enterovirus, human coronaviruses-HKU1, human coronaviruses-229E, human coronaviruses-NL63 and *Chlamydophila pneumoniae* were not detected.
ADV, adenovirus; HBoV, human bocavirus; HCoV, human coronaviruses; HMPV, human metapneumovirus; HRV, human rhinovirus; InfA, influenza A virus; MP, *Mycoplasma pneumoniae*; PIV, parainfluenza virus; RSV, respiratory syncytial virus.

1.75:1 (3202:1828) in patients who are HMPV negative (P=0.771). The median age in months of patients who are HMPV positive was 23.5 months (IQR10–36 months).

### Codetection with common respiratory pathogens in patients who are HMPV positive

We also tested HMPV-positive samples for 17 common respiratory pathogens. Of the 103 patients who are HMPV positive, 84 (81.6%) patients had only HMPV detection and 19 (18.4%) patients had HMPV plus other pathogen(s) detected. Nine out of 17 respiratory pathogens (52.9%) were detected and the most common codetection pathogens were RSV (36.8%, 7/19), HCoV-OC43 (15.8%, 3/19), MP (15.8%, 3/19) and HBoV (15.8%, 3/19) (table 1). The male to female ratio was 2:1 (56:28) in patients with only HMPV detected and 1.38:1 (11:8) in patients with HMPV plus other pathogen(s) detected (P=0.469).

### Age distribution of patients who are HMPV positive

Overall, there was a significant difference in HMPV prevalence between patients >5–14 years (0.7%, 5/734) and those ≤5 years (2.2%, 98/4399) (P=0.004). The patients were divided into six age groups: 0–3 months, >3–6 months, >6–12 months, >1–2 years, >2–5 years and >5–14 years. The distribution of HMPV prevalence between these age groups was significantly different (P=0.03), and children >1–2 years had the highest prevalence (2.8%, 21/752) (figure 1).

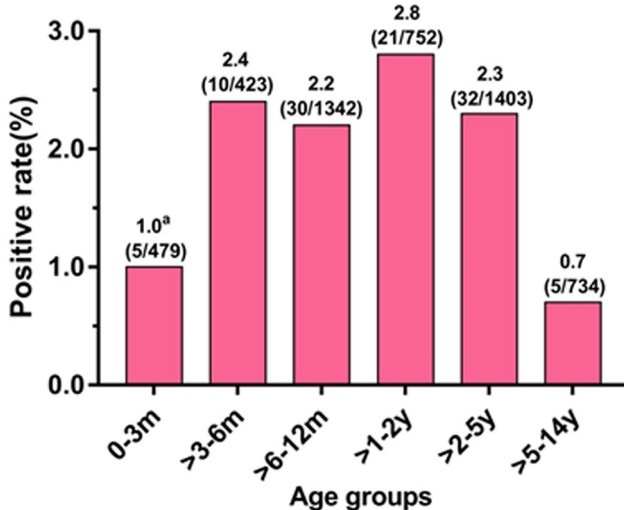

**Figure 1** Age distributions of patients with human metapneumovirus (HMPV). [a]Date were presented as HMPV positive rate (number of patients who are HMPV positive/number of patients in each age group); m: month(s); y: year(s).

### Seasonal distribution of HMPV

Over the 3-year study period, the prevalence of HMPV peaked twice every year (figure 2). Large peaks occurred in March 2014 (8%, 16/200), May 2015 (7.6%, 8/105) and February 2016 (8.7%, 9/103). Small peaks occurred in November 2014 (1.7%, 2/121), September 2015 (2.8%, 2/72) and May 2016 (1.5%, 2/135) (figure 2).

### Clinical presentation of patients who are HMPV positive

We analysed the clinical presentation of 101 of the 103 (98.1%) patients who are HMPV positive, the other two patients were excluded from this analysis because of incomplete clinical data. The main clinical features of patients who are HMPV positive included cough (100.0%, 101/101), abnormal pulmonary breath sound (91.1%, 92/101), fever (88.1%, 89/101), expectoration (77.2%, 78/101), coryza (50.5%, 51/101) and wheezing (46.5%,

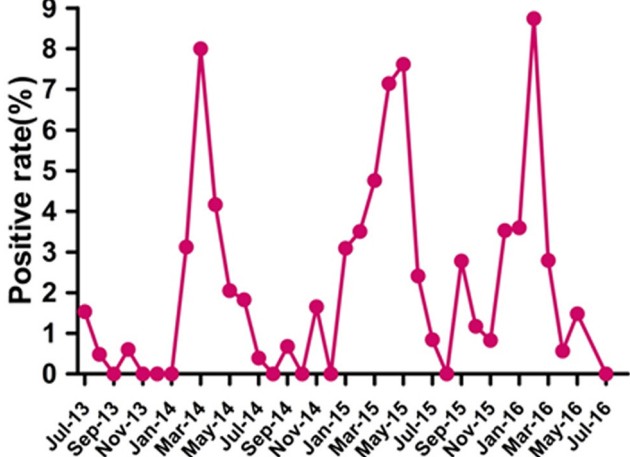

**Figure 2** Seasonal distribution of human metapneumovirus infection in paediatric patients hospitalised with acute respiratory infection from July 2013 to June 2016 in Guangzhou.

**Table 2** Clinical presentations of patients who are HMPV positive

| Diagnosis/symptom | Total HMPV (n=101) | Patients with only HMPV and HMPV plus other pathogen(s) detected. | | |
| --- | --- | --- | --- | --- |
| | | Single HMPV (n=83) | Copathogens (n=18) | P value* |
| URTI | | | | |
| Nasal obstruction | 40 (39.6) | 34 (41.0) | 6 (33.3) | 0.605 |
| Coryza | 51 (50.5) | 44 (53.0) | 7 (38.9) | 0.309 |
| Sneeze | 1 (1.0) | 1 (1.2) | 0 (0) | –† |
| Cough | 101(100) | 83(100) | 18(100) | –‡ |
| Expectoration | 78 (77.2) | 66 (79.5) | 12 (66.7) | 0.351 |
| Hoarseness | 1 (1.0) | 1 (1.2) | 0 (0) | –† |
| LRTI | | | | |
| Abnormal pulmonary breath sound§ | 92 (91.1) | 75 (90.4) | 17 (94.4) | 0.582 |
| Wheezing | 47 (46.5) | 40 (48.2) | 7 (38.9) | 0.473 |
| Anhelation | 24 (23.8) | 21 (25.3) | 3 (16.7) | 0.551 |
| Increasing lung markings¶ | 6 (7.1) | 4 (5.8) | 2 (13.3) | 0.304 |
| Bronchiolitis¶ | 8 (9.5) | 7 (10.1) | 1 (6.7) | 0.677 |
| Pneumonia¶ | 11 (13.1) | 10 (14.5) | 1 (6.7) | 0.415 |
| Bronchopneumonia¶ | 56 (66.7) | 46 (66.7) | 10 (66.7) | 0.626 |
| Systemic influenza-like symptoms | | | | |
| Fever (≥38°C) | 89 (88.1) | 76 (91.6) | 13 (72.2) | 0.037 |
| Chill | 7 (6.9) | 6 (7.2) | 1 (5.6) | 0.8 |
| Debilitation | 11 (10.9) | 11 (13.3) | 0 (0) | –† |
| Gastrointestinal illness | | | | |
| Vomiting | 21 (20.8) | 16 (19.3) | 5 (27.8) | 0.522 |
| Poor appetite | 19 (18.8) | 17 (20.5) | 2 (11.1) | 0.513 |
| Diarrhoea | 7 (7.0) | 3 (3.6) | 4 (22.2) | 0.018 |

Data are presented as n (%) of each group.
Percentages sum to >100% because some patients had more than one clinical presentations.
*Two-tailed $\chi^2$ test comparing the distribution of each illness or diagnosis between patients with only HMPV and HMPV plus other pathogen(s) detected.
†Not tested because the number of positive samples obtained was too small.
‡Not tested because the prevalence was 100%.
§Including phlegmatic, wheezing, bubbling and moist rales.
¶Diagnosed by chest radiography, and a total of 84 patients who are HMPV positive (69 patients with only HMPV detected and 15 patients with HMPV plus other pathogen(s) detected) were examined. HMPV, human metapneumovirus; LRTI, lower respiratory tract infection; URTI, upper respiratory tract infection.

47/101). Of 84 with a chest radiography, 56 (66.7%) were diagnosed with bronchopneumonia, 11 (13.1%) were pneumonia and 8 (9.5%) were bronchiolitis (table 2). We also compared the clinical features of patients with only HMPV detected and those of the patients with HMPV plus other pathogen(s) detected. Fever and diarrhoea were the two symptoms that were identified as statistically different between these two groups. A fever (≥38°C) was seen more common in patients with only HMPV detected (91.6%,76/83), compared with 72.2% (13/18) in patients with HMPV plus other pathogen(s) detected (P=0.037). Conversely, diarrhoea appeared more often in patients with HMPV plus other pathogen(s) detected (22.2%, 4/18) than in the patients with only HMPV detected (3.6%, 3/83) (P=0.018) (table 2).

## DISCUSSION

Viruses are the most frequent cause of respiratory infections.[20] HMPV has been recognised as an important cause of ARI since its discovery in 2001. A previous study reported HMPV was responsible for approximately 5%–10% of hospitalisations in children suffering from ARI,[14] creating a considerable clinical and economic burden worldwide. HMPV detection rates vary according to geographic location, and the incidence of HMPV may show seasonal or annual patterns in the same area. In previous studies,[21–30] HMPV was detected in approximately 1%–17% of ARI cases. In this 3-year study, 103 of 5133 (2.0%) patients were positive for HMPV, which is similar to previous reports of HMPV in patients with ARI in Southern China[31] and Japan.[32]

Children under 5 years of age are most susceptible to HMPV infection, and those younger than 2 years of age are at the greatest risk of developing serious conditions.[15] In our study, HMPV-positive cases appeared more frequently in patients ≤5 years old (2.2%, 98/4399) than in patients >5 years old (0.7%, 5/734) (P=0.004), which is consistent with many previous studies.[3 30 33 34] Specifically, HMPV prevalence was highest in children >1–2 years (2.8%, 21/752). This may be explained by the immunity against HMPV, which is passed down from mother to child, gradually subsiding over time. Moreover, children were more contact with outside world as they grew older. Therefore, children of this age need more attention to prevent HMPV infection. Moreover, our findings suggested no difference in risk between sex, there was no significant difference in the HMPV prevalence between male and female patients (P=0.771), which is consistent with previous studies.[35 36]

HMPV has a typical seasonal distribution. HMPV activity is largely affected by different local climate factors. The prevalence of HMPV in temperate climates peaks at the end of winter or in early spring.[37 38] In our study, seasonal peaks of HMPV were detected in February 2016, March 2014 and May 2015. This finding indicates that HMPV circulates primarily during the spring and early summer in Guangzhou, this pattern is similar to previous reports from other subtropical areas.[22 39] Furthermore, we also found small peaks in November 2014, September 2015 and May 2016, the reason for this is unknown. It should be noted that the seasonal distribution of HMPV overlaps with RSV in this geographical area.[19] Knowing the epidemiological characteristics of HMPV will help public health authorities and clinicians to improve strategies for controlling HMPV infection.

HMPV coinfection with other respiratory pathogens has been reported in many studies, including RSV,[40] InfA,[41] InfB,[42] PIV,[43] ADV,[43 44] HBoV,[45] HCoV,[43 46] HRV,[43] EV,[43] MP[41] and CP.[41] In our study, most patients who are HMPV positive only had HMPV (81.6%, 84/103), and there was no significant difference in the HMPV prevalence according to sex between the patients with only HMPV detected and the patients with HMPV plus other pathogen(s) detected (P=0.469). The codetection rate was 18.4% (19/103), and RSV (36.8%, 7/19) was the most frequently codetected pathogen, which might be because of the overlapping seasonal distribution of these two pathogens.[47] Codetection rates of HCoV-OC43, MP, HBoV, ADV, HRV, InfA, PIV2 and PIV3 were over 5%, suggesting a broad range of respiratory pathogens could coexist with HMPV in ARI paediatric patients.

HMPV most commonly causes URTI and LRTI in young children. The clinical manifestations of patients who are HMPV positive are similar to those of patients who are RSV positive, especially in young children.[48–50] In previous studies, the most frequent diagnoses of children hospitalised with HMPV infection were pneumonitis and bronchiolitis.[3 51 52] In some studies, bronchiolitis and recurrent wheezing/pneumonia were the main clinical diagnoses,[7 22 53 54] while in other studies, bronchopneumonia, bronchiolitis and bronchial asthma exacerbation were the main clinical diagnoses.[34] Our study showed that 66.7% (56/84), 9.5% (8/84) and 13.1% (11/84) of patients who are HMPV positive were diagnosed with bronchopneumonia, bronchiolitis and pneumonia by chest radiography, respectively. Of all the clinical features recorded, cough (100.0%, 101/101), abnormal pulmonary breath sound (91.1%, 92/101), fever (88.1%, 89/101), expectoration (77.2%, 78/101), coryza (50.5%, 51/101) and wheezing (46.5%, 47/101) occurred frequently in patients who are HMPV positive, similar to those reported previously.[16 29 52 55]

Most of the patients who are HMPV positive had a fever (88.1%, 89/101); it appeared more frequently in patients with only HMPV detected than in the patients with HMPV plus other pathogen(s) detected (P=0.037). Diarrhoea was not a major symptom in patients who are HMPV positive (7.0%, 7/101), but it had a higher prevalence in patients with HMPV plus other pathogen(s) detected than in the patients with only HMPV detected (P=0.018), suggesting that diarrhoea is probably caused by other pathogens. No significant difference was found in the other clinical features between the patients with only HMPV detected and the patients with HMPV plus other pathogen(s) detected, similar to the results of a previous study.[51]

In our study, RSV (36.8%, 7/19) was the most frequently codetected pathogen in patients who are HMPV positive. Whether HMPV/RSV coinfection is more severe than the respective monoinfection remains unclear. A previous study showed that children with HMPV/RSV coinfection were more likely to develop pneumonia; however, disease severity was not increased.[53] Conversely, Semple *et al*[56] reported that HMPV/RSV coinfection can cause more severe bronchiolitis in patients. In our study, no significant difference in the diagnosis of pneumonia was observed between the patients with only HMPV detected and the patients with HMPV plus other pathogen(s) detected. It has been confirmed that a history of prematurity, particular age groups, and the presence of chronic diseases increases the risk of severe LRTI among children infected with HMPV and RSV.[51] Our study suggests that the clinical manifestations of HMPV infection are complex and diverse; our data will be helpful in the diagnosis of HMPV.

In this study, we analysed the epidemiological characteristics and clinical characteristics of HMPV in >5000 children with ARI in two tertiary hospitals in Guangzhou, which is an international metropolis and the most important political, economic and cultural centre in Southern China. Therefore, the results are not limited to two hospital cases, but also represent and reflect HMPV infection in children with ARI in South China and play a positive role in the prevention and diagnosis of HMPV infection in the area.

The study had some limitations. First, collection of data on symptoms and physical findings in infants and young children require experienced medical staff, patient's

cooperation and consent from the patient's guardian. Consequently, it is possible we will have incomplete or even an inaccurate description of patients' manifestations. Second, because our study mainly focused on HMPV, other common respiratory pathogens, including bacterial pathogens, were not tested and subsequently our study does not give a fully account of respiratory pathogen infection in hospitalised paediatric patients with ARI. Third, the characteristics of patients who are HMPV negative and outcome data (discharge/death) were not analysed further, because our study only focused on the epidemiological and clinical features of HMPV positive cases. This might affect our understanding of the clinical features of patients who are HMPV positive. Despite these shortcomings, our results provide a valuable insight into the epidemiological and clinical characteristics of HMPV present in patients hospitalised with ARI in Guangzhou, Southern China.

## CONCLUSIONS

HMPV is an important respiratory pathogen in children with ARI in Guangzhou, China, particularly in children under 5 years old. In future, our data can be used by public health authorities and clinicians to improve the management of HMPV infection in children.

**Author affiliations**
[1]State Key Laboratory of Respiratory Diseases, The First Affiliated Hospital of Guangzhou Medical University, Guangzhou Medical University, Guangzhou, People's Republic of China
[2]Department of Pediatrics, The First Affiliated Hospital of Guangzhou Medical University, Guangzhou Medical University, Guangzhou, People's Republic of China
[3]Department of Pediatrics, Sun Yat-Sen Memorial Hospital, Sun Yat-Sen University, Guangzhou, People's Republic of China

**Acknowledgements** We thank the study volunteers for their generous participation. We thank Yinghua Zhou, Haiping Huang, Jing Zhang and Jing Ma for technical assistance. We thank Xiaofeng Li and Zhengshi Lin for paper polishing.

**Contributors** RZ, WL, LZ and DL designed the study. LZ, DL, WL, SQ, DX, XL and TL performed pathogen testing. DC and WT collected the clinical data. All authors participated in the data analysis. LZ, DL, RZ and WL drafted the manuscript. All authors read and approved the final version of this manuscript.

**Funding** This study was supported by The State Major Infectious Disease Research Program (RZ) (2017ZX10103011-003), National Natural Science Foundation of China (WKL) (31500143), Guangzhou Science and Technology Program key projects (RZ) (201508020252).

**Competing interests** None declared.

**Patient consent** Parental/guardian consent obtained.

**Ethics approval** The study was approved by the First Affiliated Hospital of Guangzhou Medical University Ethics Committee.

**Provenance and peer review** Not commissioned; externally peer reviewed.

**Data sharing statement** No additional data are available.

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
