## [Reviewer comments · BMJ Open]

ARTICLE DETAILS

TITLE (PROVISIONAL)	Epidemiology and clinical features of human metapneumovirus in hospitalized pediatric patients with acute respiratory illness: a cross-sectional study in southern China, from 2013 to 2016
AUTHORS	Zhang, Ling; Liu, Wenkuan; Liu, Donglan; Chen, Dehui; Tan, Weiping; Qiu, Shuyan; Xu, Duo; Li, Xiao; Liu, Tiantian; Zhou, Rong

VERSION 1 – REVIEW

REVIEWER	Kevin Zhang Southern Medical University, China
REVIEW RETURNED	15-Sep-2017

GENERAL COMMENTS	Zhang et al investigated the epidemiological and clinical features of human metapneumovirus infection in pediatric patients hospitalized with acute respiratory illness, which is important for the further diagnosis, control and prevention of infectious diseases caused by HMPV. 1. The Figure 1, "Age distributions of patients with HMPV, positive number/total number". what does the total number mean?2. The line 203 to 205, "Moreover, our findings suggested a male predominance, there was no significant difference in the HMPV positivity rate between males and females ($p=0.771$). $p > 0.05$". How to prove a male predominance?3. The line 243 to 244, suggesting that fever was the key clinical presentation caused by HMPV infection. The evidence was weak. Alternatively, the authors can change to another state.4. What is the major difference between this study and the other previous studies in HMPV epidemiology?5. The authors detected the most of common respiratory viruses by RT-PCR. However, in Results, the data were not shown in Table or Figure.6. Have the authors done any sequencing of HMPV gene to confirm the accuracy of the detection method?7. Is there any special type that the authors found from these HPMV-positive specimen ?8. p.33-34: what do ">6-12", ">1-2", ">2-5" mean? P42-43: 71.6%, (76/83) vs (72.2%, (13/18) , $p=0.037$? could the author re-calculate the numbers? P43-44, please revise this sentence. The language should be re-checked carefully.
--

REVIEWER	M.J. Groome MRC: Respiratory and Meningeal Pathogens Research Unit, University of the Witwatersrand, Johannesburg, South Africa
REVIEW RETURNED	28-Sep-2017

GENERAL COMMENTS	This study describes the clinical presentation, age distribution, and seasonality of HMPV-positive children < 14 years of age hospitalised with acute respiratory infection at two hospitals in the Guangzhou, China. While there are limited data on HMPV in low and middle income countries, there are several issues in this manuscript which need to be addressed. The case definition for ARI is unclear – you describe clinical symptoms, signs and diagnoses, some of which are very difficult to assess in very young children. It would be better to use standardised case definitions e.g. WHO. A major limitation is lack of data on HMPV-negative patients, which would act as controls. It's difficult to use clinical presentation to diagnose HMPV when you have only looked at the clinical presentation in the HMPV-positive patients. You would need to compare the clinical presentation to that in HMPV-negative patients in order to draw any meaningful conclusions. These and other comments are detailed below. Major comments  1. It is difficult to attribute causality by detection of a pathogen, especially in instances where multiple pathogens are detected. Would be better to say HMPV detection rather than HMPV infection, and refer to patients as HMPV-positive rather than HMPV-infected throughout the manuscript. 2. Are the hospitals used in the study primary, secondary or tertiary hospitals? 3. Please clarify your case definition. Were patients with pneumonia on CXR but without two of the listed symptoms included in the study? Were acute and chronic infections included? If only acute, how did you define this? What about readmissions? Did you include these? Who categorised the patients into the four groups? The attending physician? It's unclear if the attending physicians are the researchers? 4. You describe symptoms associated with HMPV infection. Symptoms are usually described by the patient whereas signs are elicited by the medical practitioner. Your description of "symptoms" includes signs e.g. abnormal pulmonary breath sounds (determined by auscultation by the physician) and diagnoses (determined by the physician e.g. bronchopneumonia). It would be better to separate out symptoms from signs and diagnoses made by the attending physician. Throughout the manuscript the word "symptoms" is used to describe signs and diagnoses. Please amend throughout, including Table 2. 5. Some of the symptoms are difficult to assess in young children e.g. pharyngeal discomfort. Your category of "influenza-like symptoms" includes chills, dizziness, headache and myalgia – how were these assessed in a young child? What is meant by "expectoration"? 6. Who gets a CXR? Some of the patients seem to have been included based on a CXR diagnosis of pneumonia. 7. You mention that samples are taken as standard of care. Who gets a swab – all children with respiratory symptoms? Only hospitalised children or outpatients as well? When is the swab taken? What pathogens are routinely tested for?
---

8. 5133 swabs were collected. Where there any children who did not have a swab collected? Any children excluded?

9. Why could data on the HMPV-negative patients not be collected? Or even on a subset of them? You state that you have a large sample size yet data were only collected on 103 HMPV-positive patients.

10. Why were only the HMPV-positive cases tested for additional pathogens? It would be more informative to have tested some of the HMPV negative samples to assess the detection rate of other pathogens in HMPV positive vs. HMPV negative patients. You state that swabs are done as standard of care – what pathogens are routinely tested for?

11. Samples testing: although you state that this has been previously described, you need to include some details in brief so that reader can ascertain what was done in your study.

12. Do you have information on the outcomes of the HMPV-positive patients? Did any die? Require ICU?

13. Was ethics approval obtained for the study?

14. Results: top of pg 10. You state that there a significant difference in the proportion of HMPV-positives by age group (line 161-162), yet in line 163-165 state that there was no difference. Please clarify.

15. Table 1, footnote: How many patients had more than 2 viruses detected?

16. Your conclusion states that this study may be helpful in the control and prevention of infectious diseases – this is a very broad statement. Perhaps you can expand on exactly how this related to HMPV. It is quite difficult to determine the aetiology of respiratory tract infections based solely on symptoms, as the clinical presentation is often very similar.

17. Discussion, pg 14: You state that “fever was the key clinical presentation caused by HMPV infection” – I don’t think that this is an accurate conclusion as you did not compare this to the proportion of HMPV-negative patients with fever. Similarly, was the proportion with diarrhoea similar in HMPV-negative patients? It is very difficult to draw conclusions when you do not have a control group.

18. Add a limitations section to your discussion.

19. Please review the grammar throughout e.g. I do not know what is meant by “...a significant difference was found between the distribution of the HMPV-positive patients...” (bottom of pg 9 and top of pg 10).

Minor comments:

1. Results: It is usually easier for the reader to understand proportions rather than ratios. Consider using proportions throughout instead of sometimes ratios and sometimes proportions.

2. As the median age is quite low, consider presenting age as months not years i.e. median age in months with IQR.

VERSION 1 – AUTHOR RESPONSE

Reviewer: 1

Reviewer Name: Kevin Zhang

Institution and Country: Southern Medical University, China

Please state any competing interests: None declared

Please leave your comments for the authors below

Zhang et al investigated the epidemiological and clinical features of human metapneumovirus infection in pediatric patients hospitalized with acute respiratory illness, which is important for the further diagnosis, control and prevention of infectious diseases caused by HMPV.

1. The Figure 1, “Age distributions of patients with HMPV, positive number/total number”. What does the total number mean?

Response:

Total number was the number of patients in the age group. To make it clear, the sentence has been revised as: “Data were presented as HMPV positive rate (number of HMPV-positive patients/number of patients in each age group)”.

2. The line 203 to 205, “Moreover, our findings suggested a male predominance, there was no significant difference in the HMPV positivity rate between males and females ($p=0.771$). $p > 0.05$ ”.

How to prove a male predominance?

Response:

Thanks for your effort to help revise this MS, there is a mistake in our writing draft. The result showed no gender difference in HMPV-infection patients, and the sentence has been revised as: “Moreover, our findings suggested no gender predominance, there was no significant difference in the HMPV positive rate between males and females ($p=0.771$).....” (line 226-228)

3. The line 243 to 244, suggesting that fever was the key clinical presentation caused by HMPV infection. The evidence was weak. Alternatively, the authors can change to another state.

Response:

The description has revised as: “Most of the HMPV-positive patients had a fever (88.1%, 89/101); it appeared in the single HMPV-positive patients more frequently than in the co-pathogens-positive patients ($p=0.037$)”. (line 266 to 268)

4. What is the major difference between this study and the other previous studies in HMPV epidemiology?

Response:

The prevalence of viruses varies widely because of factors such as geographical, climatic, crowd, social activity and so on. HMPV epidemiology in the temperate regions and in developed countries has a better description. In this work, we found that the seasonal distribution of HMPV in Guangzhou, a subtropical region, was different from the previous reports, mostly found in February 2016, March 2014, and May 2015 (Figure 2). A wider age distribution (3 months to 5 years old) was also found in this work.

5. The authors detected the most of common respiratory viruses by RT-PCR. However, in Results, the data were not shown in Table or Figure.

Response:

Our study mainly focused on HMPV, common respiratory pathogens were detected only in HMPV-positive patients (METHODS section, line 160) and the results had been shown in Table 2. It was a limitation of the study that not all samples detected for common respiratory pathogens, and discussed in line 289 to 293.

6. Have the authors done any sequencing of HMPV gene to confirm the accuracy of the detection method?

Response:

Yes, we evaluated the detection methods before a large scale usage.

7. Is there any special type that the authors found from these HPMV-positive specimen ?

Response:

It is difficult to culture HMPV in the lab; we sequenced some HMPV positive sample target for F/G gene, no special type was found up to now.

8. p.33-34: what do ">6-12", ">1-2", ">2-5" mean?

Response:

In this work, to analyze the age distribution of HMPV, the patients were divided into six age groups including 0-3 months, >3-6 months, >6-12 months, >1-2 years, >2-5 years, and >5-14 years. For example, the meaning of >3-6 months was more than three months less than or equal to six months.

P42-43: 71.6%, (76/83) vs (72.2%, (13/18) , p=0.037? could the author re-calculate the numbers?

Response:

The percentage has been corrected as "Fever ($\geq 38^{\circ}\text{C}$) (91.6%, 76/83) was detected more often in single HMPV-positive patients than in co-pathogens-positive patients (72.2%, 13/18) ($p=0.037$)" (line 42-44).

P43-44, please revise this sentence.

Response:

The sentence has been revised as "...whereas diarrhea was presented more common in co-pathogens-positive patients (22.2%, 4/18) than single HMPV-positive patients (3.6%, 3/83) ($p=0.018$)". (line 44 to 45)

The language should be re-checked carefully.

Response:

We have revised the MS and made changes to improve the quality of the MS. Thanks for your effort to help revise this MS.

Reviewer: 2

Reviewer Name: M.J. Groome

Institution and Country: MRC: Respiratory and Meningeal Pathogens Research Unit, University of the Witwatersrand, Johannesburg, South Africa

Please state any competing interests: None declared

Please leave your comments for the authors below

This study describes the clinical presentation, age distribution, and seasonality of HMPV-positive children < 14 years of age hospitalised with acute respiratory infection at two hospitals in the Guangzhou, China.

While there are limited data on HMPV in low and middle income countries, there are several issues in this manuscript which need to be addressed. The case definition for ARI is unclear – you describe clinical symptoms, signs and diagnoses, some of which are very difficult to assess in very young children. It would be better to use standardised case definitions e.g. WHO. A major limitation is lack of data on HMPV-negative patients, which would act as controls. It's difficult to use clinical presentation to diagnose HMPV when you have only looked at the clinical presentation in the HMPV-positive patients. You would need to compare the clinical presentation to that in HMPV-negative patients in order to draw any meaningful conclusions. These and other comments are detailed below.

Major comments

1. It is difficult to attribute causality by detection of a pathogen, especially in instances where multiple pathogens are detected. Would be better to say HMPV detection rather than HMPV infection, and refer to patients as HMPV-positive rather than HMPV-infected throughout the manuscript.

Response:

Thanks for your effort to help revise this MS. We have revised the descriptions and highlighted throughout the MS.

2. Are the hospitals used in the study primary, secondary or tertiary hospitals?

Response:

Two hospitals in this work are tertiary hospitals, and we add the description in line 26 and in line 91.

3. Please clarify your case definition. Were patients with pneumonia on CXR but without two of the listed symptoms included in the study? Were acute and chronic infections included? If only acute, how did you define this? What about readmissions? Did you include these? Who categorised the patients into the four groups? The attending physician? It's unclear if the attending physicians are the researchers?

Response:

ARI was defined as an illness that ARI was defined as an illness that presented with at least two of the listed symptoms or diagnosed with pneumonia by chest radiography during the previous week (line 95 to 98). So, patients with pneumonia on CXR but without two of the listed symptoms were included in the study.

Children were enrolled if they presented the symptoms or diagnosed with pneumonia during the previous week. Patients were excluded if their symptoms were not presented during this time, thus, the chronic infections were not included in this work. Patient readmission were also enrolled or excluded according to the time restriction, if the time interval was long, the patients would be included as a new case.

The attending physicians collected and categorized the patients' clinical presentations, we have described in section of "Contributors" (line 314).

4. You describe symptoms associated with HMPV infection. Symptoms are usually described by the patient whereas signs are elicited by the medical practitioner. Your description of "symptoms" includes signs e.g. abnormal pulmonary breath sounds (determined by auscultation by the physician) and diagnoses (determined by the physician e.g. bronchopneumonia). It would be better to separate out symptoms from signs and diagnoses made by the attending physician. Throughout the manuscript the word "symptoms" is used to describe signs and diagnoses. Please amend throughout, including Table 2.

Response:

We have amended the description throughout the MS. we have separated the symptoms and diagnoses in Method section (line 104; line 116 to 117), Results section (line 193 to 196), Discussion

section (line 259 to 261). We have revised the Table 2 using Diagnosis/Symptom instead of Symptom, and we have moved the three diagnostic results “Bronchiolitis”, “Pneumonia”, “bronchopneumonia” together to the bottom of the LRTI to make it easy to read.

5. Some of the symptoms are difficult to assess in young children e.g. pharyngeal discomfort. Your category of “influenza-like symptoms” includes chills, dizziness, headache and myalgia – how were these assessed in a young child? What is meant by “expectoration”?

Response:

Yes, it is very difficult to assess some symptoms in very young children. It is a limitation of the work, collection of symptoms and physical findings in infants and young children often requires the experience of medical staff, patient's cooperation, and the knowledge of patient's guardian; but it is still possible to have incomplete or even false description of patients' manifestations. We have described in the discussion section. (line 286-289).

Expectoration means producing sputum.

6. Who gets a CXR? Some of the patients seem to have been included based on a CXR diagnosis of pneumonia.

Response:

According to the patient's condition, some patients conducted chest radiography under the doctor's judgment. We have added instructions as “Chest radiography was performed based on the clinical situation of the patients” in Methods section (line 98 to 99).

7. You mention that samples are taken as standard of care. Who gets a swab – all children with respiratory symptoms? Only hospitalised children or outpatients as well? When is the swab taken? What pathogens are routinely tested for?

Response:

In this study, only hospitalized patients with ARI were studied in this work, the inclusion criteria of the patients and the sampling of the patients have been described in the part of “Study design and respiratory sample collection” of Methods section (line 93 to 95). Our study mainly focused on HMPV, and HMPV were routinely detection in this work, and other common respiratory pathogens were tested in HMPV positive patients as described in Methods section (line 134 to 142).

8. 5133 swabs were collected. Where there any children who did not have a swab collected? Any children excluded?

Response:

All enrolled patients had the swab collected. No children excluded.

9. Why could data on the HMPV-negative patients not be collected? Or even on a subset of them? You state that you have a large sample size yet data were only collected on 103 HMPV-positive patients.

Response:

Thanks for the advice, it would be better to analyze the data including the HMPV-negative patients, but in this study we focused on the features of HMPV-positive patients. The limitation has been discussed in Discussion section (line 293-296).

Totally 5133 patients were enrolled in three years in this study; it was a large sample size relatively to analyze the distribution of HMPV. The prevalence of HMPV may vary widely because of many factors such as geographical, climatic, crowd, social activity and so on, in this study, 103/5133 of patients were detected with HMPV. The results were reliable because of large sample size screening.

10. Why were only the HMPV-positive cases tested for additional pathogens? It would be more informative to have tested some of the HMPV negative samples to assess the detection rate of other pathogens in HMPV positive vs. HMPV negative patients. You state that swabs are done as standard of care – what pathogens are routinely tested for?

Response:

Thanks for the advice, in this study we focused on the features of HMPV-positive patients, HMPV were routinely detection in this work as described in Methods section (line 134 to 142). It is a large material and would be an expensive procedure of testing all HMPV-negative samples for other pathogens, and we discussed the limitation in Discussion section (line 289-293).

11. Samples testing: although you state that this has been previously described, you need to include some details in brief so that reader can ascertain what was done in your study.

Response:

We have added the description as "...In brief, 50 µl RNA were extracted from 200 µl sample, and real-time PCR was conducted using 25 µl reaction mix containing M-MLV, Taq polymerase and 5 µl extracted RNA. Cycling conditions included an initial reverse transcription at 55°C for 10 min incubation at 94°C for 2 min, followed by 40 cycles of 94°C for 10 sec and 55°C for 35 sec (ABI-7500 real-time PCR instrument, Life Technologies, Singapore)" in line 127 to 132 in Methods section.

12. Do you have information on the outcomes of the HMPV-positive patients? Did any die? Require ICU?

Response:

As far as we know, there are no deaths here, but we do not have the record whether the patients enter the ICU at last.

13. Was ethics approval obtained for the study?

Response:

Yes, ethics approval were obtained and described in line 328 to 331.

14. Results: top of pg 10. You state that there a significant difference in the proportion of HMPV-positives by age group (line 161-162), yet in line 163-165 state that there was no difference. Please clarify.

Response:

In this study, we analyzed the age distribution of HMPV among the patients of six age groups. On the whole, HMPV had age distribution differences among patients under 14 years of age ($p=0.03$). Relatively high HMPV detection rates were found among the patients with age groups of >3–6 months, >6–12 months, >1–2 years, and >2–5 years, and there was no statistical difference of HMPV distribution among these patients with upper four age groups ($p=0.865$). To make it clear, we added a sentence: "...Thus, as a whole, HMPV prevalence obviously had a differential age distribution in ARI pediatric patients, predominant in patients from 3 months to 5 years old (2.4%, 93/3920)." (line 180 to 182).

15. Table 1, footnote: How many patients had more than 2 viruses detected?

Response:

There were five patients had more than 2 pathogens detection. We have revised the Table 1, and listed all co-pathogens-positive patients in the table to make it easy to read.

16. Your conclusion states that this study may be helpful in the control and prevention of infectious diseases – this is a very broad statement. Perhaps you can expand on exactly how this related to HMPV. It is quite difficult to determine the aetiology of respiratory tract infections based solely on symptoms, as the clinical presentation is often very similar.

Response:

We have revised the statement as “HMPV is a common respiratory pathogen in children with ARI in Guangzhou, China, particularly in children from 3 months to 5 years old. HMPV epidemic has a seasonal variation. The clinical characteristics of HMPV infection has its own pattern in children aged from infants to juveniles. Bronchopneumonia is the major clinical diagnosis. In the future, our data should be taken into account when local public health authority and medical community try to manage HMPV infection in children.” (line 301-306)

17. Discussion, pg 14: You state that “fever was the key clinical presentation caused by HMPV infection” – I don’t think that this is an accurate conclusion as you did not compare this to the proportion of HMPV-negative patients with fever. Similarly, was the proportion with diarrhoea similar in HMPV-negative patients? It is very difficult to draw conclusions when you do not have a control group.

Response:

The description has revised as: “Most of the HMPV-positive patients had a fever (88.1%, 89/101); it appeared in the single HMPV-positive patients more frequently than in the co-pathogens-positive patients ($p=0.037$)”. (line 266 to 268)

As to diarrhea, we compared the manifestation of single HMPV-positive and co-pathogens-positive patients and found more frequently of diarrhea present in co-pathogens-positive patients than single HMPV-positive patients ($p=0.018$), it can reasonably infer the conclusion that diarrhea may be caused by other pathogens. To make it more clear, the sentence has been revised as “...Although diarrhea was not a major symptom in HMPV-positive patients (7.0%, 7/101), but it had a higher incidence rate in co-pathogen-positive patients than in single HMPV-positive patients ($p=0.018$), suggesting that diarrhea could be caused by other pathogens.” (line 268-271).

18. Add a limitations section to your discussion.

Response:

We have added the limitation discussion (line 286-299).

19. Please review the grammar throughout e.g. I do not know what is meant by “...a significant difference was found between the distribution of the HMPV-positive patients...” (bottom of pg 9 and top of pg 10).

Response:

The sentence has been revised as “...a significant difference of HMPV detection rate were found between patients >5–14 years (0.7%, 5/734) and those ≤ 5 years (2.2%, 98/4399) ($p=0.004$)...” (line 171-172).

We have revised the MS and made changes to improve the quality of the MS.

Minor comments:

1. Results: It is usually easier for the reader to understand proportions rather than ratios. Consider using proportions throughout instead of sometimes ratios and sometimes proportions.

Response:

We have added the proportions in discussion.

2. As the median age is quite low, consider presenting age as months not years i.e. median age in months with IQR.

Response:

We have changed the description (line 153-154, line 156-158).

Thanks for your effort to help revise this MS. We hope the revised MS can meet your requirement.

VERSION 2 – REVIEW

REVIEWER	Qiwei Zhang Southern Medical University
REVIEW RETURNED	28-Oct-2017

GENERAL COMMENTS	Most of the comments are well responded.
--

REVIEWER	M.J. Groome MRC: Respiratory and Meningeal Pathogens Research Unit, University of the Witwatersrand, Johannesburg, South Africa
REVIEW RETURNED	08-Nov-2017

GENERAL COMMENTS	Thank you for addressing my previous comments. I still feel that certain points have not yet been clarified adequately. Although you addressed the comments, you didn't always make the necessary changes in the manuscript. Other readers will likely have similar questions so you need to clarify in the manuscript itself, not just in your responses. There are still grammar errors within the manuscript which need to be addressed more thoroughly. Major comments 1. Abstract, Results: You state that HMPV was “mostly distributed in children with the age groups...” and then provide a p-value of $p=0.865$. What you did find was that HMPV was more prevalent in children <5 years compared to older children (>5-14 years), and this was a significant finding, so rather include this in the abstract as its an important finding. There wasn't a significant difference between the different age groups under 5 years.2. Abstract, Results/Main results/discussion: You still list “abnormal pulmonary breath sound” as a symptom. As I understand this would be heard by the physician on auscultation, so wouldn't be reported by the parent/patient. If you mean something different with this wording, then please change the words accordingly.3. Abstract, Results/Main results/discussion: “single HMPV-positive” and “co-pathogens-positive” patients – it is not clear what you mean and this could be better phrased. Rather use “patients with only HMPV detected” compared to “HMPV plus other pathogen(s) detected”.4. Abstract, conclusions/Conclusion: HMPV was found in 2% of children in your study. “Common” implies something which occurs often. I wouldn't conclude that it is a common pathogen. It may be an important pathogen especially in children <5years but is not necessarily a common cause.5. Case definition response: Your response must clarify the case definition within the manuscript. Mention in the text of your manuscript that patients had to have symptoms in the preceding week. Add in the text (not just in the contributors section) that the attending physicians collected the clinical data, etc.
---

6. Case definition response: what do you mean by “if the time interval was long” – you need to specify the time interval for what was considered a new case.

7. Response to comment on symptoms. I agree that collection of symptoms and physical findings in infants and young children requires experienced clinical staff. But how does a physician assess pharyngeal discomfort in a 6 month old? Or headache? Or dizziness? One cannot use adult symptoms for influenza to diagnose this in a child. You can only report symptoms that apply to young children. Please revise.

8. Response on CXR: according to your results all 101 HMPV-positive children had a CXR available. Please state this in the results before presenting the CXR data.

9. Response on samples are taken as standard of care. I do not understand why, if samples are taken as part of standard of care, you would only be testing for HMPV and not other pathogens? What was the purpose of the swab for routine care? If samples were collected only as part of your study, then it is not standard of care and you need to remove the statement.

10. Response on enrollment: so swabs were collected on all hospitalised children <14years over the 3-year period without any exceptions or non-consent from the parents? This is unusual for a study of this magnitude. Please then state that there were no refusals of consent in your study.

11. Please state as a limitation that you did not collect outcome data (discharge/death) on the patients.

12. Age distribution of HMPV-positive patients: please revise this whole paragraph. You found a significant difference between prevalence in children ≤ 5 years and those > 5 years ($p=0.004$). You found no significant difference in the different age groups in the under 5s ($p=0.965$). So you can't say that HMPV was most frequently detected in children $> 1-2$ years. Your statement “Thus, as a whole, HMPV prevalence obviously had a differential age distribution in ARI pediatric patients, predominant in patients from 3 months to 5 years old (2.4%, 93/3920),” is not supported by your data. There was no significant difference in prevalence in the children 3m-5years – previously stated with a p-value of 0.865.

13. Discussion/Conclusion responses: see comment on abstract above regarding calling HMPV a common infection.

14. Discussion/Conclusion: You conclude that “The clinical characteristics of HMPV infection has its own pattern” – to conclude this you need to be comparing the clinical characteristics to something, which your study did not do. Please remove this.

15. Discussion/Conclusion: “our data should be taken into account when local public health authority and medical community try to manage HMPV infection in children.” There is no specific treatment for HMPV available. Please revise.

	Minor comments:  1. General: you don't need to reference figures and tables again in the discussion. They are referenced in the results. 2. Strengths and limitations section: write sample size in numbers (>5000) not words. 3. Strengths and limitations section: "infancy to juveniles" – what is the definition of a juvenile, rather state your age range for clarity. 4. Results, pg 9: "In total, 5133 pediatric patients aging from infants..." – give the age of your youngest patient, not just "infants". You should be able to get the range of ages from your data – include this. 5. Line 269 – you only measured prevalence not incidence rate, please correct. 6. Line 302: the use of "epidemic" here is incorrect. How are you defining an epidemic? Else remove the term.
--	---

VERSION 2 – AUTHOR RESPONSE

Reviewer: 1

Reviewer Name: Qiwei Zhang

Institution and Country: Southern Medical University

Please state any competing interests: None declared.

Please leave your comments for the authors below
Most of the comments are well responded.

Response:

Thanks for your kind help.

Reviewer: 2

Reviewer Name: M.J. Groome

Institution and Country: MRC: Respiratory and Meningeal Pathogens Research, Unit, University of the Witwatersrand, Johannesburg, South Africa

Please state any competing interests: None declared

Please leave your comments for the authors below

Thank you for addressing my previous comments. I still feel that certain points have not yet been clarified adequately. Although you addressed the comments, you didn't always make the necessary changes in the manuscript. Other readers will likely have similar questions so you need to clarify in the manuscript itself, not just in your responses. There are still grammar errors within the manuscript which need to be addressed more thoroughly.

Major comments

1. Abstract, Results: You state that HMPV was "mostly distributed in children with the age groups..." and then provide a p-value of $p=0.865$. What you did find was that HMPV was more prevalent in children <5 years compared to older children (>5-14 years), and this was a significant finding, so rather include this in the abstract as its an important finding. There wasn't a significant difference between the different age groups under 5 years.

Response:

We have revised the ms, and changed the description as:

“HMPV was more prevalent in children ≤ 5 years (2.2%, 98/4399) compared with older children ($>5-14$ years) (0.7%, 5/734) ($p=0.004$).” (line 30).

2. Abstract, Results/Main results/discussion: You still list “abnormal pulmonary breath sound” as a symptom. As I understand this would be heard by the physician on auscultation, so wouldn't be reported by the parent/patient. If you mean something different with this wording, then please change the words accordingly.

Response:

We have revised these sentences as:

“HMPV-positive patients presented with a wide spectrum of clinical features, including cough (100.0%, 101/101), abnormal pulmonary breath sound (91.1%, 92/101),...” (line 34).

“The main clinical features of HMPV-positive patients included cough (100.0%, 101/101), abnormal pulmonary breath sound (91.1%, 92/101), ...” (line 180).

“Of all the clinical features recorded, cough (100.0%, 101/101), abnormal pulmonary breath sound (91.1%, 92/101), fever (88.1%, 89/101), expectoration (77.2%, 78/101), coryza (50.5%, 51/101),...” (line 247).

3. Abstract, Results/Main results/discussion: “single HMPV-positive” and “co-pathogens-positive” patients – it is not clear what you mean and this could be better phrased. Rather use “patients with only HMPV detected” compared to “HMPV plus other pathogen(s) detected”.

Response:

We have changed the description throughout the MS, thanks for your advice.

4. Abstract, conclusions/Conclusion: HMPV was found in 2% of children in your study. “Common” implies something which occurs often. I wouldn't conclude that it is a common pathogen. It may be an important pathogen especially in children <5 years but is not necessarily a common cause.

Response:

We have changed the description as “HMPV is an important respiratory pathogen in children with ARI ...”. (line 44 and line 289)

5. Case definition response: Your response must clarify the case definition within the manuscript. Mention in the text of your manuscript that patients had to have symptoms in the preceding week. Add in the text (not just in the contributors section) that the attending physicians collected the clinical data, etc.

Response:

We have checked the case definition and described as following: “ARI was defined as an illness that presented with at least two of the following clinical presentations: cough, nasal obstruction, coryza, sneeze, dyspnoea during the previous week. Patients, who were diagnosed with pneumonia by chest radiography during the previous week, were also included in the study, even if they did not show the clinical features described above. Some patients, who had been cured and discharged some time ago and readmitted because of new episodes of ARI, if met the recruitment criteria, were included in the study as new cases, otherwise excluded.” (line 87-93).

We have added the distribution of attending physicians in clinical data collecting as: “The patients’ clinical presentations or diagnoses were recorded from patients’ medical records by attending physicians,...” (line 99-100).

6. Case definition response: what do you mean by “if the time interval was long” – you need to specify the time interval for what was considered a new case.

Response:

Case definition has been checked and revised as following to make it clear: “ARI was defined as an illness that presented with at least two of the following clinical presentations: cough, nasal obstruction, coryza, sneeze, dyspnoea during the previous week. Patients, who were diagnosed with pneumonia by chest radiography during the previous week, were also included in the study, even if they did not show the clinical features described above. Some patients, who had been cured and discharged some time ago and readmitted because of new episodes of ARI, if met the recruitment criteria, were included in the study as new cases, otherwise excluded.” (line 87-93).

7. Response to comment on symptoms. I agree that collection of symptoms and physical findings in infants and young children requires experienced clinical staff. But how does a physician assess pharyngeal discomfort in a 6 month old? Or headache? Or dizziness? One cannot use adult symptoms for influenza to diagnose this in a child. You can only report symptoms that apply to young children. Please revise.

Response:

Thanks for your advice, pharyngeal discomfort, headache, myalgia, dizziness were excluded from the MS.

8. Response on CXR: according to your results all 101 HMPV-positive children had a CXR available. Please state this in the results before presenting the CXR data.

Response:

There might be some confusion. Case definition has been revised to make it clear (line 87-93). Not all patients had CXR. The result of CXR has been presented in MS: “..., The main diagnosis by chest radiography was bronchopneumonia (55.4%, 56/101), followed by pneumonia (10.9%, 11/101) and bronchiolitis (7.9%, 8/101).”

9. Response on samples are taken as standard of care. I do not understand why, if samples are taken as part of standard of care, you would only be testing for HMPV and not other pathogens? What was the purpose of the swab for routine care? If samples were collected only as part of your study, then it is not standard of care and you need to remove the statement.

Response:

Removed.

10. Response on enrollment: so swabs were collected on all hospitalised children <14years over the 3-year period without any exceptions or non-consent from the parents? This is unusual for a study of this magnitude. Please then state that there were no refusals of consent in your study.

Response:

The patients were enrolled according the inclusion criteria. Informed written consent was obtained from parents or legal guardians, which has been described in MS (line 316).

11. Please state as a limitation that you did not collect outcome data (discharge/death) on the patients.

Response:

We have added the description in “Strengths and limitations of this study” section (line 56-58), and discussion section (line 281-284).

12. Age distribution of HMPV-positive patients: please revise this whole paragraph. You found a significant difference between prevalence in children ≤ 5 years and those > 5 years ($p=0.004$). You found no significant difference in the different age groups in the under 5s ($p=0.965$). So you can't say that HMPV was most frequently detected in children $> 1-2$ years. Your statement “Thus, as a whole, HMPV prevalence obviously had a differential age distribution in ARI pediatric patients, predominant in patients from 3 months to 5 years old (2.4%, 93/3920),” is not supported by your data. There was no significant difference in prevalence in the children 3m-5years – previously stated with a p-value of 0.865.

Response:

We have revised this part in MS as following “Overall, there was a significant difference in HMPV prevalence between patients $> 5-14$ years (0.7%, 5/734) and those ≤ 5 years (2.2%, 98/4399) ($p=0.004$). The patients were divided into six age groups: 0–3 months, $> 3-6$ months, $> 6-12$ months, $> 1-2$ years, $> 2-5$ years, and $> 5-14$ years. The distribution of HMPV prevalence between these age groups was significantly different ($p=0.03$), and children $> 1-2$ years had the highest prevalence (2.8%, 21/752) (Figure 1).” (line 164-169).

13. Discussion/Conclusion responses: see comment on abstract above regarding calling HMPV a common infection.

Response:

We have changed the description as “HMPV is an important respiratory pathogen...”.

14. Discussion/Conclusion: You conclude that “The clinical characteristics of HMPV infection has its own pattern” – to conclude this you need to be comparing the clinical characteristics to something, which your study did not do. Please remove this.

Response:

Removed

15. Discussion/Conclusion: “our data should be taken into account when local public health authority and medical community try to manage HMPV infection in children.” There is no specific treatment for HMPV available. Please revise.

Response:

The study was done for collecting more knowledge of HMPV, and we hope to these data were useful information for future treatment and prevention of HMPV. We have revised the MS to make it clear as: “In future, our data can be used by public health authorities and clinicians to improve the management of HMPV infection in children.” (line 290-292).

Minor comments:

1. General: you don't need to reference figures and tables again in the discussion. They are referenced in the results.

Response:

Removed.

2. Strengths and limitations section: write sample size in numbers (>5000) not words.

Response:

The sentence has been revised as: “5133 patients hospitalized with acute respiratory illness at two large municipal tertiary hospitals (hospital beds > 1000) were enrolled on the study over 3 years.” (Line 49-50).

3. Strengths and limitations section: “infancy to juveniles” – what is the definition of a juvenile, rather state your age range for clarity.

Response:

The sentence has been revised as: “Patients aged from 1 day to 14 years old were tested for HMPV using Taqman real-time PCR.” (line 51-52)

4. Results, pg 9: “In total, 5133 pediatric patients aging from infants...” – give the age of your youngest patient, not just “infants”. You should be able to get the range of ages from your data – include this.

Response:

The sentence has been revised as: “In total, 5133 pediatric patients, ranging from 1 day to 14 years old were enrolled in this study.” (line 145-146)

5. Line 269 – you only measured prevalence not incidence rate, please correct.

Response:

The sentence has been corrected as: “...,but it had a higher prevalence in patients with HMPV plus other pathogen(s) detected than in the patients with only HMPV detected (p=0.018),...” (line 255-257).

6. Line 302: the use of “epidemic” here is incorrect. How are you defining an epidemic? Else remove the term.

Response:

Removed.

Thanks for your effort to review the MS.

VERSION 3 – REVIEW

REVIEWER	M.J. Groome MRC: Respiratory and Meningeal Pathogens Research Unit, University of the Witwatersrand, Johannesburg, South Africa
REVIEW RETURNED	15-Dec-2017

GENERAL COMMENTS	Thank you for addressing my comments. The manuscript has improved considerably. Just two minor comments. Minor comments: 1. Line 21: remove “was”. 2. Line 183-185 - “The main diagnosis by chest radiography was
--

	bronchopneumonia (55.4%, 56/101), followed by pneumonia (10.9%, 11/101) and bronchiolitis (7.9%, 8/101)”; and line 245-247: “Our study showed that 55.4% (56/101), 7.9% (8/101) and 10.9% (11/101) of HMPV-positive patients were diagnosed with bronchopneumonia, bronchiolitis, and pneumonia by chest radiography, respectively.” If not everyone had a chest Xray (as per your response) then the denominator can’t be 101. That was why it is confusing. The denominator must be the number who had a CXR. E.g. Of xx with a CXR, xx (xx%) were diagnosed with bronchopneumonia, etc.
--	---

VERSION 3 – AUTHOR RESPONSE

Reviewer: 2

Reviewer Name: M.J. Groome

Minor comments:

1. Line 21: remove “was”.

Response:

Thanks for your hard work, we have removed the word.

2. Line 183-185 - “The main diagnosis by chest radiography was bronchopneumonia (55.4%, 56/101), followed by pneumonia (10.9%, 11/101) and bronchiolitis (7.9%, 8/101)”; and line 245-247: “Our study showed that 55.4% (56/101), 7.9% (8/101) and 10.9% (11/101) of HMPV-positive patients were diagnosed with bronchopneumonia, bronchiolitis, and pneumonia by chest radiography, respectively.” If not everyone had a chest Xray (as per your response) then the denominator can’t be 101. That was why it is confusing. The denominator must be the number who had a CXR. E.g. Of xx with a CXR, xx (xx%) were diagnosed with bronchopneumonia, etc.

Response:

84 HMPV-positive patients (69 patients with only HMPV detected and 15 patients with HMPV plus other pathogen(s) detected) were examined with chest radiography in this work. The sentences have been revised as:

“Of 84 with a chest radiography, 56 (66.7%) were diagnosed with bronchopneumonia, 11 (13.1%) were pneumonia and 8 (9.5%) were bronchiolitis” (line 183-185);

“Our study showed that 66.7% (56/84), 9.5% (8/84) and 13.1% (11/84) of HMPV-positive patients were diagnosed with bronchopneumonia, bronchiolitis, and pneumonia by chest radiography, respectively.” (line 243-246);

Abstract and Table 2 were also revised. (line 39 and Table 2)

Response to the editorial office's requirement in mail on 22-Dec-2017

> Please ensure that your CORRESPONDING AUTHOR in your main document and Scholar One submission system are the same. If more than one author needs to share credit as first or senior author then to have a footnote saying 'xx and yy contributed equally to this paper' instead of listing two corresponding authors

Response:

In this work, Rong Zhou and Wenkuan Liu designed the study. Wenkuan Liu also performed pathogen testing and drafted the manuscript. Considering Wenkuan Liu’s contribution in the work, We listed Wenkuan Liu as co-corresponding author with Rong Zhou before this revision. However, the magazine can not allow more than one corresponding author, so here we change the list of Wenkuan

Liu as the second author in MS, and declare that "Ling Zhang, Wenkuan Liu and Donglan Liu contributed equally to this work". All authors have agreed with the current version.